

# Disentangled self-attention neural network based on information sharing for click-through rate prediction

Yingqi Wang[1,2], Huiqin Ji[1,2], Xin He[1,2], Junyang Yu[1,2], Hongyu Han[3,4], Rui Zhai[1,2] and Longge Wang[1,2]

[1] Henan University, School of Software, Kaifeng, Kaifeng, China
[2] Henan University, Henan Provincial Engineering Research Center of Intelligent Data Processing, Kaifeng, China
[3] Henan University, Henan Key Laboratory of Big Data Analysis and Processing, Kaifeng, China
[4] Henan University, School of Computer and Information Engineering, Henan University, Kaifeng, China

## ABSTRACT

With the exponential growth of network resources, recommendation systems have become successful at combating information overload. In intelligent recommendation systems, the prediction of click-through rates (CTR) plays a crucial role. Most CTR models employ a parallel network architecture to successfully capture explicit and implicit feature interactions. However, the existing models ignore two aspects. One limitation observed in most models is that they focus only on the interaction of paired term features, with no emphasis on modeling unary terms. The second issue is that most models input characteristics indiscriminately into parallel networks, resulting in network input oversharing. We propose a disentangled self-attention neural network based on information sharing (DSAN) for CTR prediction to simulate complex feature interactions. Firstly, an embedding layer transforms high-dimensional sparse features into low-dimensional dense matrices. Then, the disentangled multi-head self-attention learns the relationship between different features and is fed into a parallel network architecture. Finally, we set up a shared interaction layer to solve the problem of insufficient information sharing in parallel networks. Results from experiments conducted on two real-world datasets demonstrate that our proposed method surpasses existing methods in predictive accuracy.

# INTRODUCTION

In recent years, deep learning has been widely used in areas such as computer vision and natural language processing and has achieved remarkable popularity and application (*Aldarmaki et al., 2022*; *Tong & Wu, 2022*; *Reddy, 1976*). Deep neural networks may dynamically modify their internal weights and biases in response to changes in input data, enhancing their robustness and accuracy (*Santarsiero, Gori & Alonzo, 2019*). Deep learning becomes a valuable model for evaluating online user reaction rate issues like advertising click-through rates (CTR) based on the learning mentioned above capability.

Corresponding author
Hongyu Han, han-hongyu@henu.edu.cn

CTR prediction plays a vital role in industrial online advertising and recommendation systems (*Graepel et al., 2010*; *Lu, 2021*), and its purpose is to determine whether to recommend the item to users based on the likelihood of users clicking on the item. In CTR prediction, feature interaction is a common technique that combines different features to form new features to capture the interaction and nonlinear relationships between features. Utilizing effective methods for encoding feature interactions is often essential for enhancing the predictive accuracy of click-through rate models (*Gao et al., 2023*).

The manual selection of feature combinations for early-stage feature interaction is a time-consuming process that demands substantial human and financial resources, significantly depleting valuable assets. To address this issue, Factorization Machines (FM) (*Rendle, 2010*) represented each feature using latent factor vectors, and the pairwise feature interactions were modeled by taking the inner product of these latent vectors. The FNN (*Liu, 2023*) employed DNNs to learn and represent the intricate relationships among features, enabling them to uncover complex patterns and dependencies. The product-based neural network (PNN) (*Qu et al., 2016*) was a feature interaction model that leveraged the concept of inner product to capture interactions between features. One area of concern for both FNN and PNN models is their emphasis on high-order feature interactions, often overlooking the importance of low-order interactions. In order to better model the interaction of high-order and low-order features, Google proposed the model of Wide & Deep (WDL) (*Cheng et al., 2016*) in 2016, which combined the linear model with DNNs to improve the model generalization ability and take into account the memory ability. Deep & Cross (DCN) (*Wang et al., 2017*) and DeepFM (*Guo et al., 2017*) not only overcame the problem of focusing only on high-order feature interaction but also required no manual feature engineering.

According to literature (*Zhang et al., 2021*), CTR prediction models were classified into two types by combining explicit and implicit features of network modeling, namely parallel network architecture and stacked network architecture. DESTINE (*Xu et al., 2021*) was a stacked network architecture that calculated high-order feature interactions by stacking multiple disentangled self-attention layers. However, the stacked network architecture is essentially a linear model, which may not capture the higher-order interaction relationships between more complex features, limiting the expressive power of the model. Most current models adopt a parallel network architecture, where one network focuses on explicit feature interactions and the other on implicit feature interactions. In a parallel network architecture, explicit and implicit interactions are only combined at the final layer, with no information sharing occurring at the intermediate layers. This limitation diminishes the strength of interaction signals between them (*Chen et al., 2021*). Moreover, most models input features indiscriminately into parallel networks, resulting in excessive sharing of network inputs.

We discover two issues with existing models in the preceding description. Firstly, modeling unary terms is not considered in feature interaction but emphasizes the interaction between features. Secondly, it ignores the defect of no interaction between the layers of the parallel network architecture. To address the first problem, we introduce a disentangled self-attention layer. This layer divides the self-attention mechanism into two

parts: the paired term is used to model the specific interaction between two features, and the unary term is used to model the influence of one feature on other features (*Xu et al., 2021*; *Yin et al., 2020*). For the second problem, we propose a shared interaction layer to solve the problem of insufficient information sharing in the parallel network. We set up two modules in the shared interaction layer to enhance the interaction signals in parallel networks. One module distinguishes the feature distribution, and the other fuses the output features. Precisely, this article models the interaction between features with a single-layer disentangled self-attention mechanism. Different from the unary term in DESTINE, to better capture the difference and variation range between features, we map the unary term to different subspaces and obtain multi-head feature representations. The disentangled self-attention layer considers features from a global perspective and focuses on essential feature representations. Then, the obtained feature representation is fed into the shared interaction layer, which consists of two introduced modules. One is the decomposition module, and the other is the sharing module. The decomposition module is used to distinguish feature distribution in different networks by field control networks, and the sharing module can capture the layered interaction signals in parallel networks (*Singh et al., 2022*).

This article makes three main contributions, as follows:

- We introduce a disentangled self-attention mechanism and define paired terms and unary terms. It allows us to think about feature representation from a global perspective and focus on the crucial features.
- This article introduces two modules in the shared interaction layer to improve the interaction signals between parallel networks. One module distinguishes the feature distribution, and the other fuses the parallel network's features.
- Extensive experiments were conducted on two datasets to demonstrate the superior accuracy and lower loss rate of the proposed method compared to existing prediction methods.

## RELATED WORK

Improving click-through rate (CTR) prediction is a topic of ongoing research and interest among researchers and academics (*Singh et al., 2022*). The prediction accuracy constantly improves from the early linear regression (LR) and FM to the current DNN, DeepFM. In this section, we primarily concentrate on developing the CTR prediction model and discuss the approach for feature interaction. It also briefly introduces knowledge related to the disentangled self-attention mechanism.

### Click-through rate prediction

Predicting whether a user will click on the recommended item is an essential problem in recommendation systems (*Sangaiah et al., 2023*; *Guo et al., 2022*; *Aljunid & Huchaiah, 2022*). In CTR prediction models, commonly used algorithms include linear regression (LR), gradient boosted decision tree (GBDT), FM, FFM, DeepFM, WDL, deep Interest Network (DIN), deep interest evolution network (DIEN), and others. LR served as a linear

model primarily designed for sparse feature processing, while GBDT was a tree-based model proficient in managing non-linear features. FM, FFM, and DeepFM were models based on factorization machines that could handle high-dimensional and sparse features. WDL, DIN, and DIEN combined linear models and deep learning models to handle both low-dimensional and high-dimensional features. To further model automated feature interaction learning, the HoAFM (*Tao et al., 2020*) established intersectional features that were expressive and informative by stacking multiple cross layers. The co-action network (CAN) (*Cai et al., 2021*) was an effective CTR prediction model, which believed that there was no information sharing among the feature combinations of previous models, and thus introduced a dynamic pluggable feature interactive learning Unit Co-Action Unit, which realized the expression of feature combination information.

## Feature interaction

The efficacy of learning feature interactions has been demonstrated in the click-through rate prediction tasks. FM was proposed mainly to capture interactions between features through factorization. Subsequently, several FM variants, such as FFM (*Juan et al., 2016*), AFM (*Xiao et al., 2017*), FmFM (*Sun et al., 2021*) and FwFM (*Pan et al., 2018*) were proposed. In recent years, some researchers have modeled higher-order feature interactions. Parallel network architecture and stacked network architecture are two classic types of click-through rate prediction models. Figure 1 shows the classic models DeepFM and NFM in two network architectures. Stacked architecture models, such as NFM (*He & Chua, 2017*), DIN (*Zhou et al., 2018*), and DIEN (*Zhou et al., 2019*), are representative examples. In NFM, second-order feature interactions were stacked upon deep neural networks to model higher-order features. DIN and DIEN extracted interest representations from historical behavior and utilized attention mechanisms to model the relationship between user interests and the target item.

The parallel network architecture model captures interaction signals from explicit and implicit features and then fuses the information at the output layer. Notable models in this category include DeepFM, DCN, DeepCrossing (*Shan et al., 2016*), xDeepFM (*Lian et al., 2018*), and AutoInt (*Song et al., 2019*). In the parallel architecture model's implicit feature part, feature extraction primarily relied on deep neural networks (DNNs). DeepFM utilized the factorization machine (FM) structure for learning in the explicit feature interaction component. DCN proposed a cross-network approach to capture the interactions between features. The AutoInt model efficiently captured the non-linear relationships between features by adaptively learning the interaction weights between each pair of features.

## Disentangled self-attention mechanism

In natural language processing tasks, attention mechanisms are extensively employed, including machine translation, text summarization, question-answering systems (*Choi et al., 2016*; *Yang et al., 2016*). The attention mechanism generally assigns weights to each input item to reflect their importance in the target task (*Ali, Zhu & Zakarya, 2021*). Transformers (*Vaswani et al., 2017*) introduced a self-attention mechanism that enabled the model to focus on different positions in the input sequence simultaneously without

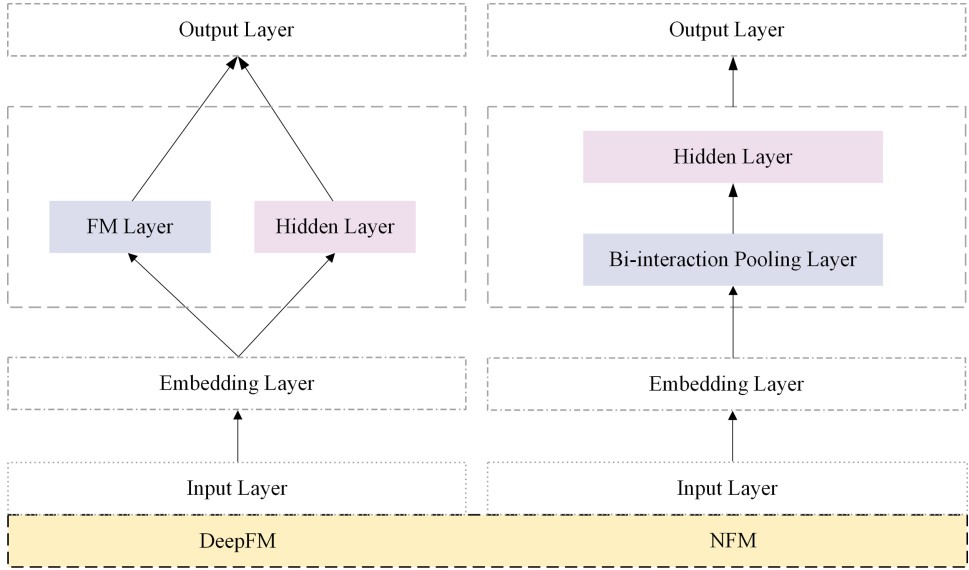

**Figure 1  Parallel architectures based on CTR prediction.**

processing them sequentially. BERT (*Devlin et al., 2018*) was built by stacking bi-directional Transformer layers and achieved good performance (*Wenzuixiong Xiong, 2023*).

The disentangled self-attention mechanism is a method for introducing decoupling based on the self-attention mechanism. It first appeared in the field of computer vision, and the two terms were used to capture the edge and the center of an image, respectively (*Yin et al., 2020*). The disentangled self-attention mechanism is divided into two parts: the paired term and the unary term. Paired terms are used to model a specific interaction between two features, and unary terms are used to model the effect of one feature on the other. DESTINE computed higher-order feature interactions by stacking multiple disentangled self-attention layers. It does not allocate the features of the unary term to distinct subspaces. In addition, we think that the stacked network architecture is essentially a linear model, which may not capture the higher-order interaction relationships between more complex features, limiting the expressive power of the model.

## METHODS

In this article, we introduce paired and unary terms in the disentangled self-attention layer, enabling us to consider the feature representation from a global perspective and focus on essential features. To address the issue of inadequate information sharing in parallel networks, we introduce two modules within the shared interaction layer. Specifically, as shown in Fig. 2, the DSAN has the following four parts: the embedding layer, the disentangled self-attention layer, the shared interaction layer, and the output layer. Firstly, the embedding layer receives the input features and converts them into compact, low-dimensional embedding vectors. Secondly, the embedded features are fed into a disentangled self-attention layer to focus on the important feature representations.

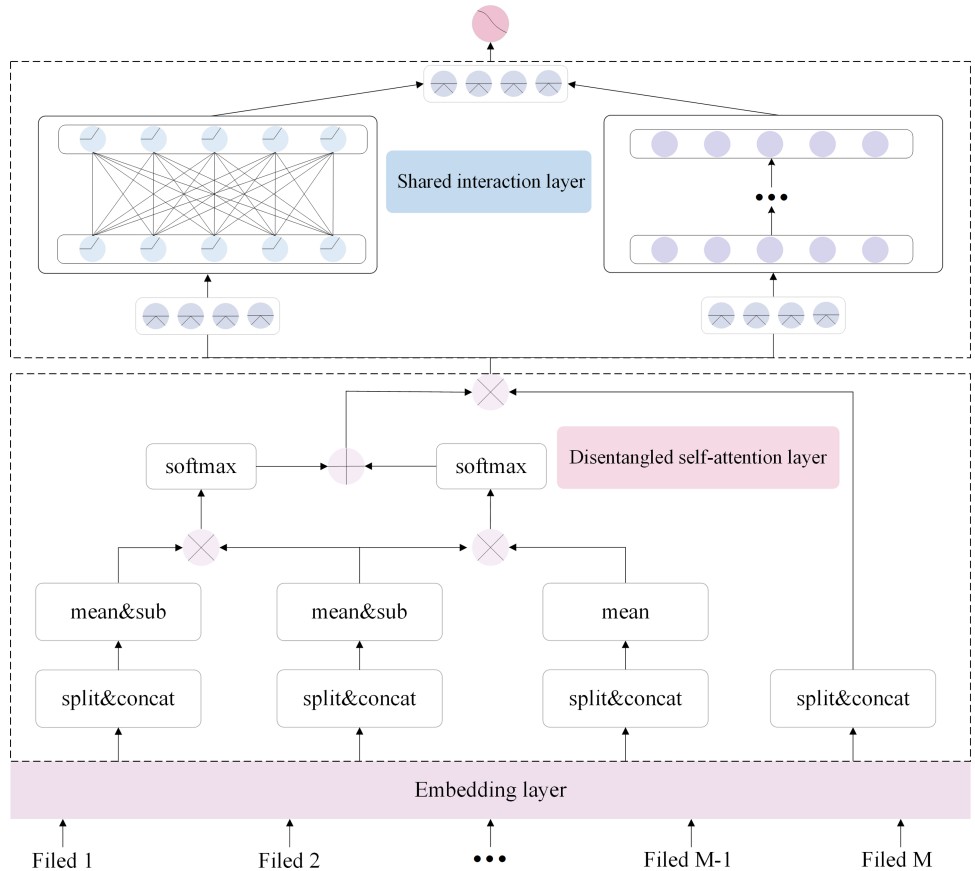

**Figure 2** **DSAN architecture based on information sharing.** The layer with dashed lines at the bottom is the disentangled self-attention layer, while the layer above it, also enclosed in a dashed box, is the shared interaction layer.

Thirdly, upon acquiring these significant representations of different features, they are channeled into the shared interaction layer to govern feature distribution. Finally, the feature representations are fed into the output layer for prediction.

## Problem definition

The main objective of CTR prediction is to assist advertisers or e-commerce companies in estimating the likelihood of users clicking on recommended items. It helps select the most valuable items for recommendation in the recommendation system, thereby maximizing its effectiveness and commercial value. Let us assume that the entire dataset $D = \{(x_1, y_1), (x_2, y_2), \ldots, (x_N, y_N)\}$ consists of $N$ examples, where each sample $x_i$ consists of $M$ user and item feature fields, and its associated label $y_i \in \{0, 1\}$ is the ground truth $i$th sample. CTR prediction is the probability of a user clicking on a recommendation using the formula $\hat{y} = f(x_i)$ under the given feature vector $x_i$.

## Embedding layer

In vision or natural language processing, the input data is often images or text signals with spatial or temporal relevance. However, input features in the recommendation system are usually sparse, and there is no apparent spatiotemporal correlation. Because CTR predictions contain discrete categorical variables that cannot be directly used for numerical computations, feature embedding is essential to forecasting click-through rates. Suppose the entire dataset consists of $N$ examples, each sample consisting of $M$ user and item feature fields. Some of these features are categorical data, while others are numerical data. The most often used approach for categorical data is feature embedding, which involves converting each sparse vector into a low-dimensional dense vector. To illustrate, if the one-hot vector of the $i$th field is designated as $x_i$, the related embedding matrix is written as $w_i$. The formulation of the $e_i$ can be represented as follows.

$$e_i = w_i x_i$$

where $x_i$ is a one-hot vector and $w_i$ is the embedding matrix of the $i$th sparse feature. By applying a conversion process, the numerical feature $x_j$ can also be transformed into the same low-dimensional space.

$$e_j = w_j x_j$$

where $w_j$ is an embedding vector and $x_j$ is a scalar value.

According to the above method, the embedding layer compresses a high-dimensional sparse vector into a low-dimensional dense vector. The equation for the embedding layer is expressed as:

$$E = \left[ e_1; e_2; \dots e_i; \dots e_j; \dots e_M \right]$$

where ";" represents the matrix stacked operation, and $M$ represents the number of feature fields.

## Disentangled self-attention layer

Self-attention is a mechanism that enables the model to assign varying degrees of importance to different positions in the input sequence, facilitating accurate predictions. In traditional self-attention, a single attention head is used to compute the attention weights between all pairs of positions in the input sequence. In multi-headed self-attention, the input sequence is mapped to multiple attention subspaces separately, and independent self-attention computations are performed in each subspace. Each attention head can learn different attention patterns and dependencies to capture multiple levels of information in the input sequence. Building upon the principles of multi-head self-attention, this article introduces the disentangled multi-head self-attention layer, which is further divided into paired terms and unary terms.

For the input feature $x_i$, it is transformed into a dense embedding vector $e_i$ by embedding search. After obtaining the low-dimensional representation of each feature, We use the dot product attention approach to simulate higher-order interactions between features. We

define four different matrices $w_q^{(h)}$, $w_{k1}^{(h)}$, $w_{k2}^{(h)}$ and $w_v^{(h)}$, then multiply $e_i$ by each of them, and the four vectors are represented as follows:

$$\begin{cases} Q_i^{(h)} = w_q^{(h)} e_i \\ K_i^{(h)} = w_{k1}^{(h)} e_i \\ \widetilde{K}_i^{(h)} = w_{k2}^{(h)} e_i \\ V_i^{(h)} = w_v^{(h)} e_i \end{cases}$$

where $Q_i^{(h)}$, $K_i^{(h)}$, $\widetilde{K}_i^{(h)}$ and $V_i^{(h)}$ are obtained by the four weights $w_q^{(h)}$, $w_{k1}^{(h)}$, $w_{k2}^{(h)}$ and $w_v \in \mathbb{R}^{d' \times d}$ respectively. $d$ is the dimension of the filed embedding, and $d'$ is the dimension of the attention.

It has been demonstrated in previous visual learning tasks that the standard self-attention mechanism is detrimental to feature learning. The disentangled self-attention mechanism is used in this article, and paired and unary terms are disentangled utilizing activation functions and embedding matrices. For the paired terms, in order to eliminate the offset information between features and enhance the expressiveness of the model, we obtain the multi-head feature representation of $Q_i^{(h)}$ and $K_i^{(h)}$, and then subtract their average values. Similar to the ordinary self-attentive mechanism, the two feature representations above are multiplied together and then passed through the softmax activation function to obtain the vector of attentional weights $P_i^{(h)}$ for the paired terms. For the unary terms, we first map the features into different subspaces, from which multiple feature representations are learned to obtain the matrix $\widetilde{K}_i^{(h)}$. Then, the mean value $\mu_{k2}^{(h)} = \frac{1}{M}\sum_{i=1}^{M} w_{k2}^{(h)} e_i = \frac{1}{M}\sum_{i=1}^{M} \widetilde{K}_i^{(h)}$ of $\widetilde{K}_i^{(h)}$ is calculated and multiplied by the matrix $K_i^{(h)} - \mu_{k1}^{(h)}$ to obtain the characteristic representations. Finally, the attentional weight vector $U_i^{(h)}$ is obtained by the softmax activation function. After we have the paired and unary terms, we add the two terms and dot them with $V_i^{(h)}$. The calculation formula is as follows:

$$\begin{cases} P_i^{(h)} = \sigma_4\left( \left(Q_i^{(h)} - \mu_q^{(h)}\right)^T \left(K_i^{(h)} - \mu_{k1}^{(h)}\right) \right) \\ U_i^{(h)} = \sigma_5\left( \mu_{k2}^{(h)} \left(K_i^{(h)} - \mu_{k1}^{(h)}\right)^T \right) \\ \text{Head}^{(h)} = \sum_{i=1}^{M} \left[ U_i^{(h)} + P_i^{(h)} \right] V_i^{(h)} \end{cases}$$

where $\sigma_4(\cdot)$ and $\sigma_5(\cdot)$ are the softmax activation function, $\mu_q^{(h)} = \frac{1}{M}\sum_{i=1}^{M} w_q^{(h)} e_i$ and $\mu_{k1}^{(h)} = \frac{1}{M}\sum_{i=1}^{M} w_{k1}^{(h)} e_i$ take average of the $Q_i^{(h)}$ and the $K_i^{(h)}$ vectors, respectively, and $M$ is the total number of features for users and items. Then, we connect all the attention heads with the following formula:

$$Z = \left[ \text{Head}^{(1)}; \text{Head}^{(2)}; \ldots; \text{Head}^{(h)} \right] w_1 + b_1$$

where $h$ is the number of attention heads, and $Z$ stacks up all the features after getting the attention head. $w_1$ means the weight matrix, and $b_1$ denotes the bias.

To preserve the information contained within the original embedded vector, a residual structure is incorporated into the network. Furthermore, normalization and ReLU

activation function procedures are implemented before the output feature representation to increase the model's stability, convergence speed, and performance.

$$L_o = \varphi(Norm(Z + w_r E)) + b_r$$

where $w_r \in \mathbb{R}^{d' \times d}$ is a linear projection matrix to avoid dimension mismatch, $\varphi(\cdot)$ is the ReLU activation function.

## Sharing interaction layer

There are two primary modules in the shared interaction layer: the decomposition module and the sharing module. The decomposition module distinguishes feature distribution in different networks by field control networks. The shared module establishes a link between the hierarchical attention and the deep network and captures the interaction signals between the two networks. These two modules are lightweight with minimal time and space complexity, and they can be easily applied to the CTR model of parallel network architecture.

*Decomposition module.* The existing CTR models provide all features equally across the two networks. The decomposition modules distinguishes features and place them in different networks in parallel architectures. Within the parallel network architecture, we employ Deep Neural Networks (DNNs) to perform implicit high-level modeling, complemented by hierarchical attention mechanisms for explicit modeling. In the aforementioned disentangled self-attention layer, we distinguish the importance of different features and generate the feature representation $L_o \in \mathbb{R}^{M \times d}$. Next, we pass $L_o$ as input into the shared interaction layer. Then, we use the decomposition module to get DNN networks and hierarchical attention networks.

$$C_l = g_i \circ L_o = \sigma_2(w_2/\gamma_2) \circ L_o$$
$$D_l = g_i' \circ L_o = \sigma_3(w_3/\gamma_3) \circ L_o$$

where $g_i$ and $g_i'$ denote the gating weight for the $i$th field, and $\circ$ denotes the Hadamard Product of two vectors. $\sigma_2$ and $\sigma_3$ are softmax activation function, $w_2$ and $w_3$ are learnable parameter, $\gamma_2$ and $\gamma_3$ are hyperparameter.

*Lian et al. (2018)* proves that each hidden layer of CrossNet is a scalar multiple of $x_0$ and interacts in a bit-wise manner, which may not capture some higher-order feature interaction patterns. In this study, the explicit feature interaction component employs hierarchical attention to build vector-wise level feature interactions, while implicit feature interaction is acquired through the fully connected layer. $C_l$ is the explicit high-order interaction part, and $D_l$ is the implicit high-order interaction part. To obtain the $l+1$-th explicit high-order feature interactions $C_{l+1}$, we aggregate the $l$th layer named $C_l$. The formula for attention aggregation is as follows:

$$C_{l+1}^j = \left( w_l^j C_l^j \circ C_0^j + b_l^j \right) + C_l^j, \quad j \in \{1,\ldots,M\}$$

where $w_l^j$ and $b_l^j$ are the weight and bias parameters on the $j$th field in the $l$th layer.

*Sharing module.* Traditional parallel network architectures independently handle explicit and implicit feature interactions, deferring information fusion to the final layer. Consequently, these traditional methods do not effectively reflect the interrelationships between parallel networks, thereby attenuating the interaction signals between explicit and implicit feature interactions. To tackle this challenge, we introduce three methods to capture the signals. We have also evaluated and compared the performance of the three methods in the experimental section.

1. We assume that $C_l$ and $D_l$ are two feature representations in a parallel network architecture. The first feature fusion, denoted as $F_{HP}$, can be expressed as $F_{HP} = C_l \circ D_l$, where $\circ$ is the Hadamard Product, and it takes their element-wise product.

2. It may not be possible to effectively model sparse feature interaction using the Hadamard Product or inner product. Therefore, we combine the inner product and the Hadamard Product to learn feature interaction. We named the second feature fusion $F_{IH}$. This interaction function is denoted as $F_{IH} = a_l^i \cdot C_l \circ D_l$, where $a_l^i$ are the shared parameters in the $l$th layer and $\cdot$ denotes the regular inner product.

3. The third approach of feature fusion concatenates two vectors. In order to keep the vector dimension of output $M \times d$, we design a feedforward layer with an activation function. The formula can be expressed as $F_{CN} = \text{ReLU}\left(w_k^T [C_l; D_l] + b_k\right)$, where $w_k$ and $b_k$ are the weight and bias parameters for the the $l$th layer, respectively.

Recurrently applying formulas in the shared interaction layer can generate an $l$th layer vector representation. The two vector representations obtained at the $l$th layer are $C_l \in \mathbb{R}^{M \times d}$ and $D_l \in \mathbb{R}^{M \times d}$ respectively. They are combined through addition before undergoing dimensional transformation. Finally, the final prediction result is obtained through a layer of linear functions.

$$\hat{y} = \sigma\left(w_c^T [C_l \oplus D_l] + b_c\right)$$

where $w_c$ and $b_c$ are parameters of weight and bias, respectively, and $\hat{y} \in (0,1)$ is the predicted label, and $\sigma$ is the activation function.

## Output layer

The loss function is utilized in the click-through rate prediction task to calculate the difference between the model forecast and the actual click-through rate situation. The model output is a probability value between 0 and 1, and the real value can only be 0 or 1. Binary cross-entropy measures the loss by calculating the difference between the predicted values and the actual values. The loss function seeks to minimize cross-entropy in the training process so that the predicted result can accurately match the actual click situation. Our loss function is $J$, which is defined as follows:

$$J = -\frac{1}{N} \sum_{i=1}^{N} \left(y_i \log(\hat{y}_i) + (1 - y_i) \log(1 - \hat{y}_i)\right)$$

where $y_i$ and $\hat{y}_i$ represent the real value and predicted value, respectively, and $y_i$ is the true label of the $i$th sample. We use a gradient descent algorithm to update model weights.

# EXPERIMENT AND ANALYSIS

In this section, we conduct experiments to verify the validity of the proposed DSAN model. Firstly, we describe the experimental settings, which include datasets, baseline models, etc. Subsequently, we compare the proposed model with the baseline model and provide a detailed explanation of the reasons behind the superior performance of our model. Finally, we perform ablation experiments to verify that each component is practical.

## Experimental settings

*Datasets.* We evaluate DSAN on two publicly available datasets, namely Criteo (https://www.kaggle.com/c/criteo-display-ad-challenge) and Avazu (https://www.kaggle.com/c/avazu-ctr-prediction). The Criteo dataset is a widely used public data set for predicting click-through rates of Internet ads. The dataset contains data on billions of advertisements displayed and clicks on anonymous websites monthly. Each data point comprises 13 digital and 26 category characteristics marked as clicked or unclicked. Avazu contains 10-day mobile ad click logs with 23 categories, including domains, types, and others. We remove the id field from the sample because it is useless in click-through rate prediction. We convert the timestamp field into hour, weekday, and is_weekend. The category characteristics of the two datasets are hashed to protect user privacy. The size and characteristics of the dataset are shown in Table 1.

*Data preprocessing.* We use the same dataset processing method as in *Zhu et al. (2021)*, dividing it into training, validation, and test sets in the ratio of 8:1:1. We treat the categorical features in the two datasets differently. For the Criteo dataset, we replace the features with less than or equal to ten with "OOV" and set the embedding dimension to 16, and then for the rare features with less than or equal to two, the embedding size is set to 40. For the Avazu dataset, we replace the infrequent features that occur less than three times with "OOV" and set the embedding dimension to 16. For the features that occur less than two times, we set the embedding size to 40. Finally, we output the preprocessed file in HDF5 format.

*Experimental details.* All models are implemented using Pytorch. All the experiments are conducted on an NVIDIA RTX 3090 GPU with 24GB of memory. The batch size for all models in both data sets is 1024. The default learning rate for all models is 0.001. We have discovered that dropout and regularization weights influence the model's performance. Therefore, we carefully adjust them between 0 and 1. In addition to this, we also fine-tuned the number of attention heads and the number of network layers of the models to get the best results. We execute an early stop strategy, stopping training when two consecutive Logloss metrics on the verification set increase. We investigate the performance of the models when the embedding dimensions are 16 and 40.

*Evaluation metrics.* In the experiment, we evaluate the performance of all methods using two famous metrics. These two metrics are widely used in CTR prediction evaluations: AUC and Logloss.

**Table 1  Statistics of datasets.**

| Dataset | #Instances | #Fields | #Features | %Positives |
|---------|-----------|---------|-----------|------------|
| Criteo | 45,840,617 | 39 | 5.55M | 26% |
| Avazu | 40,428,967 | 24 | 8.37M | 17% |

AUC: We present a quantitative evaluation of the model's performance using the ROC curve. The evaluation takes into account the sorting order of positive and negative instances. We randomly select positive and negative samples and use a trained classifier to predict both samples. The AUC measures the model's performance, with higher values indicating better performance. Suppose the data set has $Q$ positive and $G$ negative samples. The AUC formula is defined as follows:

$$AUC = \frac{\sum I\left(P_{\text{positive}}, P_{\text{negative}}\right)}{Q * G}$$

$$I\left(P_{\text{positive}}, P_{\text{negative}}\right) = \begin{cases} 1, P_{\text{positive}} > P_{\text{negative}} \\ 0.5, P_{\text{positive}} = P_{\text{negative}} \\ 0, P_{\text{positive}} < P_{\text{negative}} \end{cases}$$

Logloss: Logloss is a commonly used metric for evaluating the performance of classification models and is particularly suitable for binary classification problems. It calculates the loss based on the difference between the probability value predicted by the model and the actual label. It is defined as shown in the output layer $J$.

*Baseline models.* We compare the proposed method with state-of-the-art approaches designed explicitly for CTR tasks. Here is an introduction to the benchmark model.

- LR (*Richardson, Dominowska & Ragno, 2007*): It can only learn the first-order feature interaction, which cannot represent the interaction between features.
- FM (*Rendle, 2010*): It can effectively handle high-dimensional sparse features by modeling the interaction between features and having good generalization capabilities. But it can't simulate higher-order feature interactions.
- Wide & Deep (WDL) (*Cheng et al., 2016*): It is a hybrid architecture that combines the power of deep neural networks for learning intricate patterns with the memorization capability of a comprehensive linear model, enabling accurate click-through rate prediction by capturing both generalization and specific feature interactions.
- NFM (*He & Chua, 2017*): It is a neural network-based extension of Factorization Machines that leverages deep learning techniques to capture feature interactions and improve click-through rate prediction in various applications.
- Deep & Cross (DCN) (*Wang et al., 2017*): It is a neural network architecture incorporating cross-network operations to capture high-order feature interactions (*He et al., 2016*), enabling accurate click-through rate prediction by balancing depth-wise representation learning and explicit feature interactions.
- DeepFM (*Guo et al., 2017*): It is a hybrid approach combining deep neural networks and factorization machines, leveraging their complementary strengths to capture high-order

feature interactions and low-rank representations, enabling accurate click-through rate prediction in large-scale recommendation systems.

- xDeepFM (*Lian et al., 2018*): It is an advanced deep learning architecture integrating cross-network operations and deep neural networks, effectively capturing intricate feature interactions and hierarchical representations.
- FiBiNet (*Huang, Zhang & Zhang, 2019*): It introduces two modules: Bililinear Feature Interaction and SENet. SENet is a powerful mechanism that selectively recalibrates feature representations by learning channel-wise attention weights. The bilinear-interaction layer performs element-wise product and linear transformation operations to capture intricate feature interactions.
- InterHAt (*Li et al., 2020*): It incorporates hierarchical self-attention mechanisms to capture feature interactions at different levels, improving click-through rate prediction by effectively modeling the importance and dependencies among features in recommendation systems.
- DESTINE (*Xu et al., 2021*): It stacks multiple disentangled self-attention mechanisms to model the interaction of higher-order features, which decouples the unary feature importance calculation from the second-order feature interactions.
- DeepLight (*Deng et al., 2021*): It introduces a parallel network architecture that combines DNN and FwFM and conducts analysis compared to DeepFM and xDeepFM. This approach aims to address the challenges of increased service latency and high memory usage when delivering real-time services in production.
- CowClip (*Zheng et al., 2022*): It develops the adaptive column-wise clipping to address the model's training speed, reducing the training time from 12 h to 10 min on a single V100 GPU.
- FinalMLP (*Mao et al., 2023*): It proposes seamlessly integrating feature gating and interaction aggregation layers into an upgraded dual-stream MLP model. In other words, by combining these two MLPs, it can achieve improved performance.

## Performance comparison

In this section, we report the model's performance on two datasets. The DSAN and the best baseline are emphasized in bold and underlined formats. The "Previously Reported" column shows the best results for both datasets that we found in our existing work. The "#Params" indicates the number of parameters used in each model. Table 2 presents the performance of models, and we can draw the following conclusions:

- In the "Previously Reported" column, the performance of InterHAt is inferior to LR on both datasets. It could be attributed to differences in how the models preprocess the datasets and the splitting ratios. This article evaluates all models using the same evaluation protocol and preprocessing methods for result comparability. In addition, we perform a paired $t$-test to verify the statistical significance of the relative improvement of DSAN.
- The performance obtained on Avazu datasets is essentially the same or even higher than the reported results. For example, InterHAt and DCN receive better results on Avazu datasets. The experimental results show that LR and FM shallow models perform worse

**Table 2   Performance comparison of different models on Criteo and Avazu.** The statistical significance for each pair between our proposed model and the baseline is $p < 0.05$.

| Year | Model | Previously Reported | | Criteo(dim=16) | | | Criteo(dim=40) | | |
|------|-------|------|---------|------|---------|---------|------|---------|---------|
| | | AUC | LogLoss | AUC | LogLoss | #Params | AUC | LogLoss | #Params |
| 2007 | LR | 0.7858 | 0.4474 | 0.7901 | 0.4699 | 0.9M | 0.7902 | 0.4697 | 5.5M |
| 2010 | FM | 0.7933 | 0.4464 | 0.7903 | 0.4633 | 15.5M | 0.7902 | 0.4635 | 227.5M |
| 2016 | WDL | 0.8062 | 0.4453 | 0.8062 | 0.4457 | 15.8M | 0.8061 | 0.4459 | 230.4M |
| 2017 | NFM | 0.7968 | 0.4537 | 0.8010 | 0.4502 | 16.0M | 0.8008 | 0.4506 | 228.6M |
| 2017 | DCN | 0.8009 | 0.4425 | 0.8071 | 0.4445 | 14.6M | 0.8068 | 0.4449 | 227.4M |
| 2017 | DeepFM | 0.8085 | 0.4445 | 0.7974 | 0.4551 | 15.5M | 0.7975 | 0.4551 | 228.1M |
| 2018 | xDeepFM | 0.8091 | 0.4418 | 0.8072 | 0.4448 | 15.8M | 0.8084 | 0.4440 | 228.0M |
| 2019 | FiBiNet | 0.8103 | 0.4423 | 0.8060 | 0.4459 | 15.9M | 0.8057 | 0.4461 | 230.0M |
| 2020 | InterHAt | 0.7845 | 0.4577 | 0.8016 | 0.4497 | 14.6M | 0.8013 | 0.4499 | 222.8M |
| 2021 | DESTINE | 0.8087 | 0.4425 | 0.8083 | 0.4432 | 15.2M | 0.8081 | 0.4430 | 226.5M |
| 2021 | DeepLight | 0.8116 | 0.4403 | 0.8092 | 0.4423 | 14.6M | 0.8089 | 0.4427 | 227.5M |
| 2022 | CowClip | 0.8097 | – | 0.8090 | 0.4426 | 14.8M | 0.8087 | 0.4429 | 227.6M |
| 2023 | FinalMLP | 0.8149 | – | 0.8073 | 0.4442 | 14.7M | 0.8071 | 0.4446 | 228.0M |
| - | **DSAN** | – | – | **0.8105** | **0.4401** | 15.9M | **0.8094** | **0.4423** | 228.9M |

| Year | Model | Previously Reported | | Avazu(dim=16) | | | Avazu(dim=40) | | |
|------|-------|------|---------|------|---------|---------|------|---------|---------|
| | | AUC | LogLoss | AUC | LogLoss | #Params | AUC | LogLoss | #Params |
| 2007 | LR | 0.7676 | 0.3868 | 0.7676 | 0.3871 | 3.8M | 0.7679 | 0.3865 | 8.4M |
| 2010 | FM | 0.7793 | 0.3740 | 0.7828 | 0.3787 | 63.8M | 0.7854 | 0.3773 | 343.3M |
| 2016 | WDL | 0.7749 | 0.3744 | 0.7889 | 0.3746 | 64.1M | 0.7890 | 0.3742 | 345.4M |
| 2017 | NFM | 0.7708 | 0.376 | 0.7843 | 0.3774 | 64.3M | 0.7878 | 0.3754 | 343.8M |
| 2017 | DCN | 0.7681 | 0.3721 | 0.7876 | 0.3753 | 60.1M | 0.7901 | 0.3740 | 335.0M |
| 2017 | DeepFM | 0.7836 | 0.3742 | 0.7877 | 0.3755 | 63.8M | 0.7898 | 0.3744 | 343.4M |
| 2018 | xDeepFM | 0.7855 | 0.3737 | 0.7903 | 0.3739 | 64.0M | 0.7917 | 0.3732 | 343.9M |
| 2019 | FiBiNet | 0.7832 | 0.3786 | 0.7853 | 0.3769 | 63.9M | 0.7856 | 0.3767 | 343.6M |
| 2020 | InterHAt | 0.7582 | 0.3910 | 0.7834 | 0.3779 | 60.0M | 0.7886 | 0.3749 | 335.0M |
| 2021 | DESTINE | 0.7831 | 0.3789 | 0.7902 | 0.3738 | 60.5M | 0.7910 | 0.3733 | 335.7M |
| 2021 | DeepLight | 0.7893 | 0.3751 | 0.7890 | 0.3744 | 60.0M | 0.7893 | 0.3750 | 335.1M |
| 2022 | CowClip | – | – | 0.7906 | 0.3731 | 60.3M | 0.7911 | 0.3738 | 336.1M |
| 2023 | FinalMLP | 0.7666 | – | 0.7845 | 0.3771 | 60.1M | 0.7849 | 0.3771 | 335.2M |
| - | **DSAN** | – | – | **0.7916** | **0.3729** | 60.8M | **0.7931** | **0.3722** | 337.5M |

because they cannot catch complex higher-order feature interactions. Deep models are more advantageous in capturing complex higher-order feature interactions than shallow models and usually perform better. Models like WDL, NFM, and DeepFM, among others in deep learning, demonstrate superior performance compared to shallow models. By adjusting the parameters, we find that the differences between most models become minimal. The discrepancies between DCN and DeepFM on the Avazu dataset are negligible. During the model training process, we have noticed that most models exhibit a phenomenon called one-epoch overfitting, meaning that training for just one epoch typically suffices to achieve optimal performance. According to the relevant

literature (*Zhang et al., 2022*), it points out that feature sparsity is the cause of one epoch. Improving model performance by improving data sparsity may be a worthwhile research topic.

- The performance of the Criteo dataset is generally in line with the reported results, but in some aspects, it is slightly lower than expected. The lack of detailed hyperparameter information provided in the article resulted in us not obtaining the best combination of hyperparameter tuning and feature engineering. However, InterHAt and NFM models demonstrate superior performance on Criteo datasets, underscoring the effectiveness of our approach to data preprocessing and hyperparameter optimization. We can also observe that the Criteo dataset has better feature representation at an embedding dimension of 16, while the Avazu dataset obtains better results at higher embedding dimensions. This difference may stem from the complexity and characteristics of the datasets themselves, leading to variations in the optimal choice of embedding size.

- Tuning model parameters is one of the crucial steps in deep learning. We adjust the models regarding embedding size, number of attention heads, and network layers. We tune the embedding size to { 16, 24, 32, 40, 48 } for both datasets to investigate the effect of embedding dimensions on the model. The number of network layers in deep learning is tuned between 0 and 4. To further illustrate the need for tuning, Table 3 shows the results of the three benchmark models before and after tuning. The "Reported" reflects the findings of the existing research; "Rerun" denotes the outcome pre-adjustment; and "Retuned" signifies the outcome post-adjustment. We can observe that while DeepLight and FinalMLP didn't achieve the reported performance on the Criteo dataset, they slightly surpassed the previous performance on the Avazu dataset. This variance might arise from the distinct characteristics and complex relationships among features in these datasets. It indicates that the characteristics of the data influence a model's performance across different datasets, and it requires specific adjustments tailored to each dataset to achieve better performance.

## Performance analysis

In this section, we examine the performance of the model. We compare the proposed disentangled self-attention layer with the traditional self-attention approach and investigate the performance of three variants of the sharing module.

### *Disentangled self-attention layer performance analysis*

The disentangled self-attention mechanisms are a method that introduces decoupling based on self-attention mechanisms. We devise two comparative approaches to assess the efficacy of the disentangled self-attention mechanism proposed in this article, and we compare it with the disentangled self-attention layer in DESTINE. Method one, DSAN alt SA, combines the common self-attention mechanism with the shared interaction layer. The second method, DSAN alt DS, combines the disentangled self-attention mechanism proposed in *Xu et al. (2021)* with the shared interaction layer.

According to the experimental results in Fig. 3, our method DSAN outperforms the two methods above on both datasets. Compared with DSAN alt SA, the AUC of this

**Table 3  Results before and after hyperparameter tuning.**

| Model | Setting | Criteo | | Avazu | |
|---|---|---|---|---|---|
| | | AUC | LogLoss | AUC | LogLoss |
| InterHAt | Reported | 0.7845 | 0.4577 | 0.7582 | 0.3910 |
| | Rerun | 0.7961 | 0.4528 | 0.7618 | 0.3857 |
| | Retuned | 0.8016 | 0.4497 | 0.7886 | 0.3749 |
| DeepLight | Reported | 0.8116 | 0.4403 | 0.7893 | 0.3751 |
| | Rerun | 0.8087 | 0.4430 | 0.7887 | 0.3759 |
| | Retuned | 0.8092 | 0.4423 | 0.7893 | 0.3750 |
| FinalMLP | Reported | 0.8149 | – | 0.7666 | – |
| | Rerun | 0.8069 | 0.4461 | 0.7762 | 0.4016 |
| | Retuned | 0.8073 | 0.4442 | 0.7849 | 0.3771 |

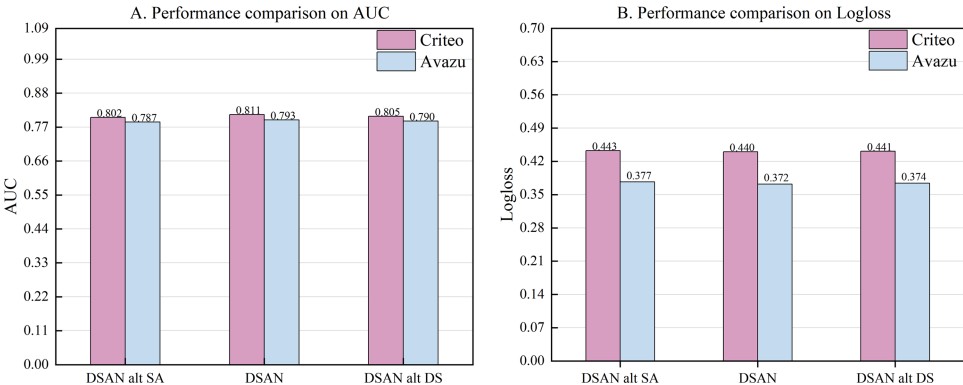

**Figure 3   (A–B) Performance comparison of three self-attention mechanisms.**

article improved by 0.9% and 0.6% on the Criteo and Avazu datasets, respectively. The results show that the disentangled self-attention mechanism proposed in this article is superior to the common self-attention mechanism. Compared with DSAN alt DS, DSAN improved the AUC by 0.6% and 0.3% on the two datasets, Criteo and Avazu, respectively. This result indicates differences in the design of disentangled self-attention layers between DESTINE and this article. In this article, we successfully capture the differences and variation range between the unary term characteristics of the disentangled self-attention layer, thus improving the model's performance.

### Different variant of sharing module

We investigate the effects of three different variants in the shared module. To capture the signal between different networks of the parallel architecture, we express the Hadamard Product of two vectors as DSAN-HP, which is also the way used in the article, and thus as DSAN. The inner product and Hadamard Product feature fusion are denoted as DSAN-IH, and the way the two vectors are connected and passed through the feed-forward neural network is denoted as DSAN-CN. As shown in Fig. 4, DSAN performs the best. It is due to

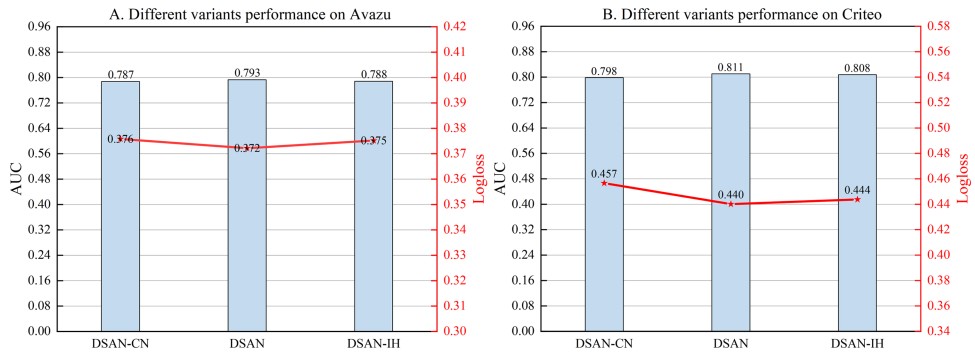

**Figure 4** (A–B) Performance comparison of different variants.

the Hadamard Product being in the same position as the elements and not involving any weights or coefficients.

## Ablation study

We conduct experiments to verify the efficacy of individual components within the DSAN model and understand their respective significance. Each experiment entails the removal of a single component while keeping the remaining components unchanged, ensuring a focused evaluation of their relative importance. The DSAN w/o DA denotes the removal of disentangled multi-head self-attention, implying only a shared interaction layer in the model. DSAN w/o PT is the removal of paired terms in disentangled multi-head self-attention, and DSAN w/o UT is the removal of unary terms in disentangled multi-head self-attention. DSAN w/o SI indicates the removal of the part of the two modules that share the interaction layer, which means that only the DNN network and the attention aggregation exist in the parallel structure.

Based on Table 4, it is evident that eliminating any component from DSAN results in a decline in model performance. Compared to DSAN w/o DA, DSAN improves the AUC values by 1.2% and 0.9% on both the Criteo and Avazu datasets, respectively. This progress demonstrates the efficacy of the introduced disentangled self-attention layer in bolstering the model's accuracy. To verify the effectiveness of the unary and pairwise terms presented in the disentangled self-attention layer, we design DSAN w/o PT and DSAN w/o UT. Compared to the DSAN w/o SI, the DSAN model exhibited 1.2% and 0.6% improvements in AUC values on the Criteo and Avazu datasets, respectively. This demonstrates that integrating decomposition and sharing modules into a parallel network architecture is effective.

## CONCLUSION

Click-through rate prediction plays an important role in recommender systems. However, the existing models ignore two aspects. One limitation observed in most models is that they focus only on the interaction of paired term features, with no emphasis on modeling unary terms. The second issue is that most models input characteristics indiscriminately

**Table 4** Ablation experiments.

| Model | Criteo | | Avazu | |
|---|---|---|---|---|
| | AUC | Logloss | AUC | Logloss |
| DSAN w/o DA | 0.7988 | 0.4581 | 0.7843 | 0.3773 |
| DSAN w/o PT | 0.8012 | 0.4436 | 0.7906 | 0.3745 |
| DSAN w/o UT | 0.7964 | 0.4597 | 0.7881 | 0.3769 |
| DSAN w/o SI | 0.7989 | 0.4589 | 0.7872 | 0.3756 |
| DSAN | 0.8105 | 0.4401 | 0.7931 | 0.3722 |

into parallel networks, resulting in network input oversharing. In this work, we propose the DSAN model for click-through rate prediction. DSAN uses the disentangled self-attention layer to learn the ambiguity of feature interactions and then inputs into the shared interaction layer. We designed two modules in the shared interaction layer. One module distinguishes the feature distribution, and the other fuses the parallel network's features. Our proposed model achieves better off-line AUC and Logloss than other models.

How to construct an interpretable click-through rate prediction network will be investigated in future work. Furthermore, we observed a one-epoch phenomenon related to click-through rate prediction during the experiment. Addressing this in future research could enhance the performance of click-through rate prediction.

# ACKNOWLEDGEMENTS

I would like to thank the anonymous reviewers whose comments and suggestions helped improve this manuscript.

## Funding

The research received funding from the Key Research and Promotion Projects of Henan Province under Grant Agreement No (222102210034, 222102210178, 222102210229 and 232102210031), and the Key Research Projects of Henan Higher Education Institutions under Grant Agreement No 22A520020. The funders had no role in study design, data collection and analysis, decision to publish, or preparation of the manuscript.

## Grant Disclosures

The following grant information was disclosed by the authors:
The Key Research and Promotion Projects of Henan Province: 222102210034, 222102210178, 222102210229, 232102210031.
The Key Research Projects of Henan Higher Education Institutions: 22A520020.

## Competing Interests

The authors declare there are no competing interests.

## Author Contributions

- Yingqi Wang conceived and designed the experiments, authored or reviewed drafts of the article, and approved the final draft.
- Huiqin Ji conceived and designed the experiments, performed the experiments, analyzed the data, performed the computation work, authored or reviewed drafts of the article, and approved the final draft.
- Xin He analyzed the data, prepared figures and/or tables, and approved the final draft.
- Junyang Yu performed the experiments, prepared figures and/or tables, and approved the final draft.
- Hongyu Han conceived and designed the experiments, prepared figures and/or tables, and approved the final draft.
- Rui Zhai performed the experiments, prepared figures and/or tables, and approved the final draft.
- Longge Wang analyzed the data, authored or reviewed drafts of the article, and approved the final draft.

## Data Availability

The data and models are available at Zenodo: haha. (2023). jihuiqin2/interaction_ctr: v0.1 (interaction). Zenodo. https://doi.org/10.5281/zenodo.8119714.

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
