# Peer review of "Disentangled self-attention neural network based on information sharing for click-through rate prediction"

_PeerJ Computer Science, doi:10.7717/peerj-cs.1764_

## Round 0.1 · original submission · Major Revisions

Dear authors: Your paper looks of good technical quality. However, it is similar to https://arxiv.org/pdf/2101.03654.pdf in title, organization, algorithm, data, and design of experiments. Your submission cites this previous paper, but does not discuss it. Please revise your submission to include a careful comparison with https://arxiv.org/pdf/2101.03654.pdf, especially its methods and their accuracy. Thank you.

---

## Round 0.2 · Major Revisions

It has been difficult to obtain rigorous reviews for this paper, so I am reluctantly making a decision based on just the opinion of Reviewer 1 (Reviewer 2's review is hardly useful). Please revise the paper thoroughly in two important ways:

1. "to claim that the improvements are significant, it is important to carry out a statistic significance test, which is missing from the article. Hence, I highly suggest that the authors conduct a statistical analysis test. The proposed model is compared to 10 baseline models. Each of these models is briefly mentioned in page 9. In most cases no references are provided. It is important that each of the methods used as baseline for comparison is correctly cited."

2. Please find all the recent papers from ICML, KDD, ICLR, and NeurIPS that report accuracy on the Criteo or Avazu datasets. For each such paper, compare your results carefully. Explain precisely how the training and test sets are identical or different. Explain whether the comparison is fair: have both methods been allowed similar hyperparameter optimization and/or other refinements?

Reviewer 1 ·

Basic reporting

This work proposes a disentangled self-attention neural network based on information sharing (DSAN) for click-through rate (CTR) prediction in recommendation systems. The DSAN model addresses limitations in existing models by considering both paired term features and unary terms in feature interactions, and by efficiently sharing information among input characteristics through a shared interaction layer.

Experimental design

According to the authors, results on Criteo and Avazu datasets demonstrate that the proposed DSAN model significantly improves AUC values and decreases Logloss values. However, to claim that the improvements are significant, it is important to carry out a statistic significance test, which is missing from the article. Hence, I highly suggest that the authors conduct a statistical analysis test.
The proposed model is compared to 10 baseline models. Each of these models is briefly mentioned in page 9. In most cases no references are provided. It is important that each of the methods used as baseline for comparison is correctly cited.

Why are the LR and FM baselines missing in the comparison reported on table 3?

It is not clear how the different hyperparameters were selected for the proposed model.

Validity of the findings

The proposed model appears to be superior due to its ability to consider both paired term features and unary terms in feature interactions, as well as it’s efficient information sharing through a shared interaction layer among input characteristics. However, no analysis was conducted to assess the individual effects of each aspect on the results. I suggest that the authors consider conducting an analysis to quantify the impact of each of these aspects on the overall performance improvements observed. This could provide valuable insights into the specific contributions of each element and further validate the effectiveness of the proposed model. Unfortunately, the ablation study does not address this with sufficient level of detail.

The manuscript is missing a discussion and conclusion section. It abruptly ends with an ablation study, but no general summary or closure is provided for the findings and their implications. This would help better appreciate the significance of the results, and their alignment with the research objectives.

Additional comments

Some typos were identified:
Figure 1: Embeedding -> Embedding
Line 140: Transformer -> Transformers
Figure 2: split & cancat - > split & concat

Reviewer 2 ·

Basic reporting

(1) The authors have revised the manuscript to address the major concerns.
(2) So, whether the revised version can be accepted will be based on the votes of group reviewers.
(3) My comment is accept with minor suggestion. This manuscript could be polished again by a fluent English speaker.
(4) It would be better to open the source code for further research in this area

Experimental design

.

Validity of the findings

.

---

## Round 0.3 · Minor Revisions

Thank you for the careful improvements. However, please search more thoroughly for recent papers with results for comparison. This is Comment 2: "Please find all the recent papers from ICML, KDD, ICLR, and NeurIPS that report accuracy on the Criteo or Avazu datasets. For each such paper, compare your results carefully. Explain precisely how the training and test sets are identical or different. Explain whether the comparison is fair: have both methods been allowed similar hyperparameter optimization and/or other refinements?"

Here are some papers that I found with a quick search:

Page 23 of https://openreview.net/pdf?id=PUaSVhBzem reports 79.05% accuracy for Criteo, better than in the current submission.

Page 927 of https://dl.acm.org/doi/pdf/10.1145/3437963.3441727 reports AUC 0.8116 for Criteo, also better than in the current submission.

The last page of https://arxiv.org/pdf/2006.15939v1.pdf has AUC 79.91% for Criteo, also better.

Page 7 of https://arxiv.org/pdf/2204.06240v3.pdf has many AUC results over 80%.

I found these 4 papers in less than 15 minutes of searching. Please find more, and answer the questions above, to make comparisons be useful.

---

## Round 0.4 · Minor Revisions

Thank you for the detailed response to my previous comments. Thank you also for finding additional relevant experimental results, especially those in the BarsCTR paper.

However, it is not enough to discuss the additional relevant papers in the response. They must all be discussed carefully in the manuscript. The BarsCTR paper is not even cited in the latest version!

“The accuracy of the three models, DeepLight, CowClip, and FLEN, is not reasonably consistent with those described in the original paper, which could be attributed to data processing and the experimental environment.” This means that those papers are scientifically invalid. The authors of this submission must justify carefully why their own results are valid.

Specifically: (1) follow all the details in the BarsCTR paper, combined with explicit critical thinking and writing about whether those details are reasonable.

(2) Remove all the current tables and figures from the submission. Replace them with similar new tables, based on new experiments that are consistent with Tables 3 and 4 of the BarsCTR paper.

Smaller issues:

The statement “The Criteo dataset size increased by 840617, and the Avazu dataset size increased by 428967 compared to those in the original manuscript” is not acceptable. Table 1 in the manuscript is not clear whether 45.0M means 45,000,000 exactly, or 45 million approximately. In any case, all experiments should use the exact same dataset versions used in the BarsCTR paper.

“On the Criteo dataset, these models perform better using an embedding dimension of 16, while on the Avazu dataset, the best performance is obtained with an embedding size of 32. It is because the Criteo dataset is relatively large, and thus, a smaller embedding size helps maintain an efficient computation speed, improving the models' performance.” This is nonsense. “Performance” is an ambiguous word. It means accuracy in “these models perform better” but “computation speed” is a completely different meaning of “performance.”

The title is capitalized incorrectly on the cover sheet.

---

## Round 0.5 · accepted · Accept

Thank you for the quick and thorough revision.

The new Table 2 is good. For clarity, I suggest writing “Previously Reported” not “Best Reported”. In line 338, write “The “Previously Reported Best Report” column shows the best results for both datasets that we found in our literature search existing work.”

Describe explicitly what the “one epoch” problem is.

Typo before 275: It should be “0.5, Ppositive = Pnegative”